# Structural Changes of the Trinuclear Copper Center in Bilirubin Oxidase upon Reduction

**DOI:** 10.3390/molecules24010076

**Published:** 2018-12-26

**Authors:** Takaki Tokiwa, Mitsuo Shoji, Vladimir Sladek, Naoki Shibata, Yoshiki Higuchi, Kunishige Kataoka, Takeshi Sakurai, Yasuteru Shigeta, Fuminori Misaizu

**Affiliations:** 1Department of Chemistry, Graduate School of Science, Tohoku University, 6-3 Aoba, Aramaki, Aoba-ku, Sendai, Miyagi 980-8578, Japan; misaizu@m.tohoku.ac.jp; 2Department of Physics, Graduate School of Pure and Applied Sciences, University of Tsukuba, 1-1-1 Tennodai, Tsukuba, Ibaraki 305-8571, Japan; 3Center for Computational Sciences, University of Tsukuba, 1-1-1 Tennodai, Tsukuba, Ibaraki 305-8577, Japan; shigeta@ccs.tsukuba.ac.jp; 4Institute of Chemistry, Centre for Glycomics, Slovak Academy of Sciences, Dubravska cesta 9, 84538 Bratislava, Slovakia; chemvlsl@savba.sk; 5Graduate School of Life Science, University of Hyogo, 3-2-1 Koto, Kamigori-cho, Ako-gun, Hyogo 678-1297, Japan; shibach@sci.u-hyogo.ac.jp (N.S.); hig@sci.u-hyogo.ac.jp (Y.H.); 6Graduate School of Natural Science & Technology, Kanazawa University, Kakuma, Kanazawa, Ishikawa 920-1192, Japan; kataoka@se.kanazawa-u.ac.jp (K.K.); tasakura@staff.kanazawa-u.ac.jp (T.S.)

**Keywords:** catalytic intermediate, protonation, Cu oxidation state, X-ray reduction, SONO, molecular orbital analysis

## Abstract

Geometric and electronic structure changes in the copper (Cu) centers in bilirubin oxidase (BOD) upon a four-electron reduction were investigated by quantum mechanics/molecular mechanics (QM/MM) calculations. For the QM region, the unrestricted density functional theory (UDFT) method was adopted for the open-shell system. We found new candidates of the native intermediate (NI, intermediate II) and the resting oxidized (RO) states, i.e., NI^H+^ and RO_0_. Elongations of the Cu-Cu atomic distances for the trinuclear Cu center (TNC) and very small structural changes around the type I Cu (T1Cu) were calculated as the results of a four-electron reduction. The QM/MM optimized structures are in good agreement with recent high-resolution X-ray structures. As the structural change in the TNC upon reduction was revealed to be the change in the size of the triangle spanned by the three Cu atoms of TNC, we introduced a new index (*l*) to characterize the specific structural change. Not only the wild-type, but also the M467Q, which mutates the amino acid residue coordinating T1Cu, were precisely analyzed in terms of their molecular orbital levels, and the optimized redox potential of T1Cu was theoretically reconfirmed.

## 1. Introduction

Bilirubin oxidase (BOD) catalyzes the bilirubin oxidation using the reduction of dioxygen into water: 2 bilirubin + O_2_
→ 2 biliverdin + 2 H_2_O. BOD belongs to the family of multicopper oxidases (MCOs) which contain unique Cu atoms, i.e., type I copper (T1Cu), type II copper (T2Cu) and type III copper (T3Cu). These Cu atoms have been classified based on their spectroscopic and magnetic properties [1]. BOD is a monomeric protein with a molecular mass of 60 kDa and consists of three domains [2,3]. The O_2_ reduction is carried out at the trinuclear Cu center (TNC), which is composed of one T2Cu and a pair of T3Cu atoms, T3aCu and T3bCu. T1Cu plays an essential role in electron transfer from the substrate to TNC. The pair of T3Cu ions in TNC of BOD is connected to T1Cu via His458-Cys457 and Cys457-His459 amino acids arranged in a Y shape, and the Cu centers are about 13 Å apart. In the same MCO family, there are ascorbate oxidase, ceruloplasmin, laccase, as well as BOD. The BODs have been widely utilized for the diagnostic analysis of bilirubin in serum. Recently, BODs have attracted attention for use as an enzymatic catalyst for the cathode of biofuel cells because of the high activity at neutral pH [4,5].

During the catalytic cycle of BOD, some intermediate states are formed [6,7] (Figure 1). In the fully oxidized state, two forms of the TNC have been characterized: the native intermediate (NI, intermediate II [8]) and the resting oxidized (RO) states. In the NI state, both the μ^3^-oxo and μ^2^-OH bridges connect the T3Cu atoms, while in the RO state, only the μ^2^-OH bridge remains between the T3Cu ions. As the transition rate from NI to RO is slower than the turnover rate of the catalytic cycle, NI is the actual catalytic center relevant to the oxidized form [9]. After a four-electron reduction, the fully reduced state (FR) is formed which contains all the Cu(I)s, and the OH ligand coordinated to T2Cu in the oxidized form is protonated to form H_2_O. O_2_ can then bind to the unsaturated coordination sites in TNC, and a two-electron reduced intermediate state, called the peroxy intermediate (PI, intermediate I [8]), is formed. After an additional two-electron reduction, the NI state is then regenerated.

We are particularly interested in characterizing the structural changes upon the four-electron reduction processes NI→FR and RO→FR within the BOD catalytic cycle. These processes are expected to be deeply related to the entire catalytic cycle and relative redox potential requirements for the T1Cu and TNC. We expected that these rules and functional features would provide important insights into the BOD catalytic activities and applications of BOD.

Recently, we reported that the wild-type (WT) BOD possesses a characteristic cross-link between Trp396 and His398, which directly coordinates to the T1Cu [10]. Met467 is another T1Cu coordinating residue that can be mutated to alter the redox potential of T1Cu. In the M467Q mutant, the Trp396-His398 cross-link is lost [10] and the catalytic activity becomes quite low (~0.3%) [11,12]. Therefore, we intend to examine theoretically how the mutation influences the electronic structures of T1Cu and TNC. In the present study, the quantum mechanics/molecular mechanics (QM/MM) method was employed to elucidate the geometrical changes and electronic states of T1Cu and TNC. For the QM region, the unrestricted hybrid density functional theory was adapted to describe the open-shell system (Figure 2). We have demonstrated that the QM/MM approach is handy and powerful enough to be used for open shell systems with transition metal centers [13,14,15,16]. Therefore, we adapted the QM/MM methodology with a large QM region, and the electronic structure analyses based on the natural orbitals, atomic spin densities and molecular orbitals have been performed in the present study.

## 2. Results and Discussions

### 2.1. T1Cu

The T1Cu is coordinated by four amino acid residues, His398, Cys457, His462, and Met467. In the oxidized state such as RO_0_, the QM/MM optimized T1Cu(II)-S(Met467) distance is *R*(T1Cu(II), S(Met467)) = 3.23 Å, which is longer compared to the other coordination distances; *R*(T1Cu(II), N(His398)) = 2.01 Å, *R*(T1Cu(II), N(His462)) = 1.98 Å, and *R*(T1Cu(II), S(Cys457)) = 2.17 Å. The coordination geometry of T1Cu is trigonal–pyramidal, where the atoms in the two His and Cys residues form an equatorial triangle and the S atom in Met467 defines the axial coordination. As shown in Figure 3a, the localized singly occupied natural orbital (LSONO) [17] for the T1Cu is composed of the Cu d_x2-y2_ orbital and 2p orbital of the Cys457 S atom. The calculated Löwdin atomic spin densities (*ρ*(S(Cys457)) = 0.53 and *ρ*(T1Cu) = 0.34) also show a large spin/unpaired electron orbital distribution over the T1Cu and S of Cys457 (Table 1).

In the reduced state, such as the FR state, T1Cu becomes Cu(I), and the atomic spin densities are not observed over the T1Cu and S of Cys457. The coordination distances around the T1Cu slightly increased as *R*(T1Cu(I), S(Met467)) = 3.44 (+0.21) Å, *R*(T1Cu(I), N(His398)) = 2.19 (+0.18) Å, *R*(T1Cu(I), N(His462)) = 2.00 (+0.02) Å, and *R*(T1Cu(I), S(Cys457)) = 2.20 (+0.03) Å, in which the distance changes upon the reduction are shown in parentheses. These very small structural changes match to minimize the reorganization energy during the electron transfer via the T1Cu site. These small structural changes can be attributed to the large spin-orbital delocalization from the d orbital of T1Cu to the p orbital of S of Cyc457.

The corresponding coordination distances around the T1Cu in the X-ray structure (PDBID = 6IQZ) [10] are *R*(T1Cu, S(Met467)) = 3.36 (−0.08) Å, *R*(T1Cu, N(His398)) = 2.06 (−0.13) Å, *R*(T1Cu, N(His462)) = 2.05 (+0.05) Å, and *R*(T1Cu, S(Cys457)) = 2.16 (−0.04) Å, in which the differences to the QM/MM optimized structure in the T1Cu(I) state are given in parentheses. These results indicated that these coordination distances are properly reproduced by the present QM/MM calculations within 0.1 Å.

### 2.2. T2Cu and T3Cu

The T2Cu is coordinated to two His residues, His94 and His401, and one water (W) or hydroxo (OH) ligand. In the states from the NI to RO_0_, the μ^3^-oxo atom additionally coordinates to TNC, and T2Cu is coordinated by the four ligands in a square planar orientation. On the other hand, the T3Cu atoms are both coordinated to three His residues and the μ^3^-oxo atom, and these ligands form a tetrahedral coordination. As shown in Figure 1, the NI, RO_1_ and FR_W_ states contain μ^2^-hydroxyl/water bridges between the pair of T3Cu ions. In the RO_1_, FR_W_ and FR states, there are no directly bridging ligands between the T2Cu and T3Cu atoms. In all these states, three Cu atoms of TNC are arranged in a triangle, which will be suitable for the four-electron reduction of O_2_ to occur at the central position during the FR to NI transition. In the oxidized states (NI and RO), all the T1Cu, T2Cu and T3Cu ions are expected to be Cu(II). In the fully reduced states (FR_W_ and FR), all the Cu ions are expected to be Cu(I). Two states with and without the μ^2^-water between two T3Cu ions were examined. These states are called FR_W_ and FR, respectively.

In the NI state, the calculated Löwdin atomic spin densities are *ρ*(T1Cu) = 0.00, *ρ*(S(Cys457)) = 0.00 and *ρ*(μ^3^-oxo) = 1.07 (Table 1). This result indicated that TNC in the NI state is one more oxidized with the μ^3^-oxo radical, while the T1Cu is one-electron reduced with Cu(I). The use of different DFT exchange-correlation functionals, such as BLYP and BHLYP, does not affect/improve the one-electron transfer from TNC to T1Cu. This means that T1Cu(I) cannot reduce the TNC, and this NI state is unstable. Therefore, other protonation states during the NI to RO transition are examined and compared. As illustrated in Figure 4, there are two intermediate states after the first and second protonations for NI, and they are called NI^H+^ and RO_W_, respectively.

For the first protonated state, NI^H+^, there are two possible protonation sites at the OH of T2Cu or at the μ^2^-OH between the T3Cu ions. The former NI^H+^ state, which is illustrated in Figure 4, is more stable than the latter one by 34.1 kcal mol^−1^. Thus, the OH of T2Cu is protonated first, and μ^2^-OH of the T3Cu ions is then protonated to RO_W_ in the subsequent protonation as illustrated in Figure 4. The energy difference for the first and second protonations is ΔΔ*E* = (Δ*E*(NI) − Δ*E*(NI^H+^)) − (Δ*E*(NI^H+^) − Δ*E*(RO_W_)) = 41.8 kcal mol^−1^. This indicated that the second protonation step is more difficult. This result is consistent with the reaction rate via RO of *k* = 0.05 s^−1^ and this reaction is very slow compared to the catalytic reaction of *k* = 350 s^−1^, in which the direct route is adapted without forming RO [6]. As the energy barrier expected for the NI to RO transition corresponds to 8.8–13.9 kcal mol^−1^, we can estimate that the first protonation step to NI^H+^ can spontaneously occur. Another possibility is that the deprotonation of the water ligand of T2Cu did not occur during the PI state formation.

The atomic spin densities at NI^H+^ are *ρ*(T1Cu) = 0.13, *ρ*(S(Cys457)) = 0.30 and *ρ*(μ^3^-oxo) = 0.75, *ρ*(Oμ^2^−OH)) = 0.29. These atomic spin densities at the T1Cu indicate that T1Cu is in a half-reduced-like Cu(1.5) oxidation state. It also suggests that T1Cu can reduce the TNC when T1Cu is further reduced. Based on these QM/MM results, we expect that NI^H+^ is one candidate for the stable native intermediate.

The RO_W_ generated after the second protonation contains a water molecule which is almost dissociated from the T3aCu (*R*(T3aCu, O_W_) = 2.58 Å). Therefore, after the release of the water molecule, the RO state will be formed easily. In the RO_W_ state, T1Cu is almost in the Cu(II) state based on the atomic spin densities (*ρ*(T1Cu) = 0.30, *ρ*(S(Cys457)) = 0.55, *ρ*(μ^3^-oxo) = 0.76, *ρ*(O(μ^2^-W)) = 0.08). These results mean that TNC in the RO_W_ state becomes very oxidative after the two-step protonation.

For the next RO state, two possible states with μ^3^-OH or with μ^3^-oxo were examined. We found that the μ^3^-oxo form is more stable than the μ^3^-OH form by 24.9 kcal mol^−1^. Therefore, the former state with μ^3^-OH, called RO_0_ should be the proper RO state compared to the latter state with μ^3^-oxo, called RO_1_, at the present QM/MM theoretical level. The Löwdin atomic densities are *ρ*(T1Cu) = 0.31 (0.32), *ρ*(T2Cu) = 0.62 (0.50), *ρ*(T3aCu) = 0.53 (0.56), and *ρ*(T3bCu) = 0.65 (0.60) for RO_0_ (RO_1_). Thus, all the T1-, T2- and T3Cu atoms in the RO states become close to the Cu(II) formal oxidation states. The localized singly-occupied natural orbitals (LSONOs) [17] for the d orbitals of TNC are shown in Figure 3b–d.

As the calculated Löwdin atomic spin densities on the Cu atoms are zero in both FR and FR_W_ states, the Cu atoms can be considered as Cu(I). Structural changes upon the reduction are mainly observed for the Cu positions, and the triangle formed by the three Cu atoms in TNC is enlarged due to the removal of the central μ^3^-oxo (Table 2). The distances become *R*(T2Cu-T3aCu) = 4.34 Å (+0.86), *R*(T2Cu-T3bCu) = 4.52 Å (+1.04), and *R*(T3aCu-T3bCu) = 4.79 Å (+1.52) in FR_W_, where the distance changes compared to the RO_0_ are shown in parentheses. After the elimination of the μ^2^-water between the T3Cu atoms, the Cu-Cu distances slightly shrink to 4.27, 4.45, and 4.60 Å, respectively. Therefore, the μ^2^-water contributes slightly to elongating the T3Cu atoms. The binding energy of the μ^2^-water is calculated to be 26.5 kcal mol^−1^. The solvation energy of the water molecule is much higher compared to the binding energy. Then the μ^2^-water will be easily released to the solvent.

The characteristic structural change in TNC upon reduction is mainly attributed to the loss of the μ^3^-oxo in the oxidized forms (NI-RO). Therefore, we propose that the percentage of the reduction of the TNC will be proportional to the square root of the area spanned by the T2Cu and T3Cu atoms.

The area of a triangle with three side lengths (*l*_12_, *l*_23_, *l*_31_) is easily calculated using Heron’s formula.
(1)$$S_{\mathrm{triangle}}=\sqrt{s\left(s-l_{12}\right)\left(s-l_{23}\right)\left(s-l_{31}\right)}$$
(2)$$s=(l_{12}+l_{23}+l_{31})/2$$

One side length *l* assumed in an ideal equilateral triangle is calculated as
(3)$$l=\sqrt{\frac{4S}{\sqrt{3}}}$$

This equation can be derived from
(4)$$S=\frac{\sqrt{3}}{4}\;l^{2}$$

As the *l* value becomes close to the actual Cu–Cu lengths, this index *l* is very meaningful and useful to characterize the structural features of TNC.

For the oxidized states such as the NI, NI^H+^, RO_W_ and RO_0_ states, the *l* values are 3.4–3.5 Å. For the reduced states, such as the FR_W_ and FR states, the *l* values are 4.4–4.5 Å (Table 3). The *l* values for the X-ray structures of MCOs are in the range of 4.3–4.4 Å except for the 3.5 Å of PDBID = 3UAA. The 3UAA is the only structure which processes the μ^3^-oxo, and it is solved as an oxidized state. Based on the root mean square deviation (RMSD), most of the X-ray structures (PDBIDs = 2XLL [3], 6IQZ [10], 3UAE [18]) are closest to the FR_W_ state except for the 3UAA structure [18], which is closest to the NI^H+^ state. Therefore, the index *l* is a very useful parameter which represents the oxidation state of the TNC. It should be noted that the index *l* will be valuable to estimate the percentage of the reduced state of the X-ray crystal structures by interpolating between *l*_ox_ = 3.382 Å and *l*_red_ = 4.538 Å, in which we assume that the oxidized (OX) and reduced (red) states are presented by the NI^H+^ and FR_W_ states, respectively.
(5)$$l=C_{ox}\;l_{ox}+(1-C_{ox})l_{red}$$
(6)$$C_{ox}=\frac{l_{red}-l}{l_{red}-l_{ox}}$$

Using Equation (6), the TNCs of 3UAA and 3UAE are estimated to be 92% and 12% oxidized states, respectively. The X-ray structures of 3UAA and 3UAE were taken under the low and high X-ray dose conditions for the Cu efflux Oxidase (CueO), respectively, and the latter data are more reduced by the X-ray exposure [18]. Therefore, the estimation using Equation (6) provides a reasonable prediction for the oxidation level of TNC.

The X-ray structure of 3UAE shows a very low electron density around the μ^2^-O atom position, which was assigned to the oxygen atom of μ^2^-water in the FR_W_ state. Based on the RMSD values, the structure of 3UAE most closely resembles the optimized structure in the FRW state which agrees with the past experimental assignment [18]. For PDBID = 6IQZ, the TNC is closest to the FR_W_ state based on the RMSD values. This assignment is appropriate because the electron density around the μ^2^-O atom position was observed, which limits the candidates to RO_1_ and FR_W_. Based on the present theoretical study, we showed that RO_1_ is very unstable compared to RO_0_. Therefore, the TNC of 6IQZ is assigned not to RO_1_ but to FR_W_.

### 2.3. Molecular Orbital Analysis for the Reduction Process of WT and M467Q BOD

Changes in molecular orbitals for the Cu centers were analyzed and the orbital energy levels are shown in Figure 5. In the oxidized state of NI, the β highest occupied molecular orbital (βHOMO) is mainly the p orbital of the μ^3^-oxo, and the β lowest unoccupied molecular orbital (βLUMO) is mainly localized on the T1Cu and S of Cys457, which is consistent with the spin density distribution. After the first protonation, in the NI^H+^ state, the βHOMO and βLUMO are mixed together, and the orbital energies of the βHOMO, βLUMO and higher βLUMOs for T2Cu and T3Cu ions become more stabilized. The higher βLUMOs are highly delocalized over the TNC. The orbital energy levels in Figure 5 clearly show that T1Cu is more easily reduced than the TNC. After the second protonation to RO_0_, the LUMOs become further stabilized and the Cu centers become more reducible. The orbital energy level for the T1Cu (βLUMO) is lowered from −0.726 eV to −2.981 eV, i.e., by −2.255 eV during the NI^H+^ to RO_0_ state transition (All the numerical data for these orbitals are summarized in the Appendix A). The average orbital energies for the TNC higher βLUMOs become 2.922 eV, 1.232 eV and −1.425 eV in the NI, NI^H+^ and RO_0_ states, respectively, i.e., they are stabilized by −1.689 eV and −2.658 eV by the protonation steps. The βLUMOs for the Cu atoms are shown in Figure 6.

In the reduced states of FR and FR_W_, the d orbitals for T1Cu and TNC are all occupied; the βHOMO corresponds to the d orbital of T1Cu and the lower βHOMOs contain the d orbital components for TNC. The orbital energies for FR and FR_W_ indicate that the β^2^-water coordination does not perturb these βHOMOs. In FR and FR_W_, the βHOMO of the T1Cu lies slightly above or almost level with the βHOMO-1 of T3Cu. This means that the redox potential of the T1Cu of the WT is maximized to reduce the TNC.

The corresponding orbital energies of the M467Q mutant are all increased. For example, in RO_0_, the βLUMO for T1Cu and the average for the TNC higher βLUMOs are −1.927 eV and −1.003 eV, which are increased by 1.053 eV and 0.422 eV, respectively. In FR_W_, the βHOMO for T1Cu and the average for the TNC lower βHOMOs are 2.099 eV and 1.340 eV, which are increased by 1.378 eV and 0.383 eV, respectively. Therefore, a consequence of the mutation is the increase of the T1Cu as well as the TNC orbital energies. The upshift of the T1Cu orbital energy level in the M467Q mutant is consistent with the decrease in the T1Cu redox potential from 470 mV to 270 mV in the M467Q mutant [12], which contributes to slowing down the electron transfer from bilirubin to T1Cu. It should be noted again that the lower shift of the T1Cu orbital is not permitted because it must be higher than the orbital levels of TNC. In this sense, the redox potential of T1Cu is optimized in the WT BOD to catalyze the bilirubin oxidation.

## 3. Materials and Methods

### Computational Details

The QM/MM calculations were performed using the NWChem program package (version 6.3) (Pacific Northwest National Laboratory, Richland, WA, USA) [19]. The unrestricted density functional theory with the B3LYP exchange-correlation functional [20] and the Grimme’s D3 dispersion correction (UB3LYP-D3) [21] were used for the QM region, and the rest of the MM region was applied to the AMBER-99 force field [22]. Basis sets used for the copper and the other atoms were LanL2DZ (Los Alamos ECP plus DZ) and 6-31G(d), respectively [23,24,25,26]. The initial coordinates of the WT BOD (PDBID: 6IQZ) and M467Q mutant (PDBID: 6IQY) were taken from their high-resolution X-ray structures determined at 1.46 and 1.60 Å resolutions, respectively [10]. The BODs from fungus *Myrothecium verrucaria* were produced in *Pichia pastoris* as recombinant proteins. The PROPKA program (GitHub, San Francisco, CA, USA) [27,28] was employed to estimate the protonation states of the titratable residues (Asp, Glu, His, Lys, and Arg). The BODs were neutralized by 12 Na^+^ ions and were solvated by MM water molecules in a sphere of 90 Å diameter (Figure 2a). First, energy minimization for the hydrogen atom was performed at the MM level. Next, a 10 ps MD simulation at 298.15 K was performed to relax all the hydrogen atoms, solvent water molecules and Na ions. After another 10 ps cooling from 298.15 K to 0 K, a 1000-step energy minimization was performed under the same constraint condition. Therefore, coordinates of the heavy atoms are the same in the X-ray structures and all the atoms added to construct the theoretical model were redistributed properly. For the non-bonded QM–MM interactions, electron embedding schemes with a 9 Å cut-off and with no cut-off were used for the geometric optimizations and energy minimization calculations, respectively. The QM region contains the T1Cu, T2Cu and 2 T3Cu atoms, 3 water molecules, and the 13 nearest amino acid residues (His94, His96, His134, His136, Trp396, His398, His401, His403, His456, Cys457, His458, His462, and Met467 in WT BOD/Gln467 in M467Q BOD) (Figure 2b,c). Especially, S of Cys457 was assumed to be deprotonated. The hydrogen link atom approach was used for treating the QM/MM boundary. The entire model contains about 38,000 atoms (about 10,000 molecules), while the QM region contains about 210 atoms. As we adapted a large QM region, the electronic structure of the QM/MM model is accurate enough by minimizing the artificial effect from the hydrogen link atoms. The molecular structures and molecular orbitals shown in the figures were drawn using PyMOL (DeLano Scientific LLC, Palo Alto, CA, USA.) [29,30] and the visual molecular dynamics (VMD) program (Theoretical and Computational Biophysics Group, Urbana, IL, USA) [31]. Complete numerical data for the QM/MM optimized structures and orbital energies are available in the Appendix A.

## 4. Conclusions

The catalytic intermediates of BOD: NI, NI^H+^, RO_W_, RO, FR_W_ and FR, were investigated using the QM/MM method. The characteristic structural changes for the T1Cu and TNC are discussed for each intermediate state, and their spin distributions and the molecular orbital levels were precisely analyzed. In the NI state, the T1Cu(I) and excessively oxidized TNC state were calculated, which is very high in energy. Therefore, one protonated state, NI^H+^, should be the proper native intermediate state. For the RO state, two protonation states with μ^3^-oxo or μ^2^-OH were examined. The former state (RO_0_), in which T3Cus and T2Cu are coordinated by μ^3^-oxo and water, respectively, is more stable than the latter (RO_1_). In the RO_0_, all the four Cu atoms are Cu(II), and their related molecular orbitals in the βLUMOs are clearly depicted in Figure 6. After the four-electron reduction, all the Cu centers in the FR_W_ and FR states become Cu(I). In Figure 7, the summary of the catalytic intermediates of BOD based on the present QM/MM calculations is depicted. Compared to the catalytic cycle in Figure 1, one additional protonation step seems to have appeared during the PI to NI transition. However, in Figure 1, one deprotonation from the water coordinating to T2Cu and one protonation to the μ^2^-oxo are necessary. These processes are not explicitly shown for clarity. Therefore, the catalytic cycle of Figure 7 indicates that the deprotonation of the T2Cu-water is not necessary and that it corresponds to a simpler reaction mechanism.

One remaining and undetermined issue for the catalytic cycle is that FR may be replaced by FR_W_ since all the X-ray structures in the fully reduced states are close to FR_W_. Because the structural differences between FR and FR_W_ are very small, it is not easy to differentiate them except for the direct observation of the μ^2^-O electron density by X-ray diffraction.

During the reduction, the Cu triangle area of the TNC increases with the loss of the μ^3^-oxo. We introduced a new index l to characterize the structural change. On the other hand, for T1Cu, very limited structural changes were calculated around T1Cu, which relates to the highly delocalized spin-orbital over the p orbital on the S atom of Cys457.

The electronic structural changes for the M467Q mutation were related to the molecular orbitals, and we showed that the energy of d orbital for the T1Cu was raised by ~1.0 eV in RO_0_. Since the d orbitals for TNC increased by ~0.4 eV, the relative increase in the orbital level of T1Cu was estimated to be ~0.6 eV. Increases in the Cu molecular orbital levels were observed for the FR states. As the orbital level of the T1Cu in FR_W_ lies only slightly above the TNC orbitals, the redox potential of T1Cu in WT BOD is maximized as the T1Cu can reduce the TNC.

## Figures and Tables

**Figure 1 molecules-24-00076-f001:**
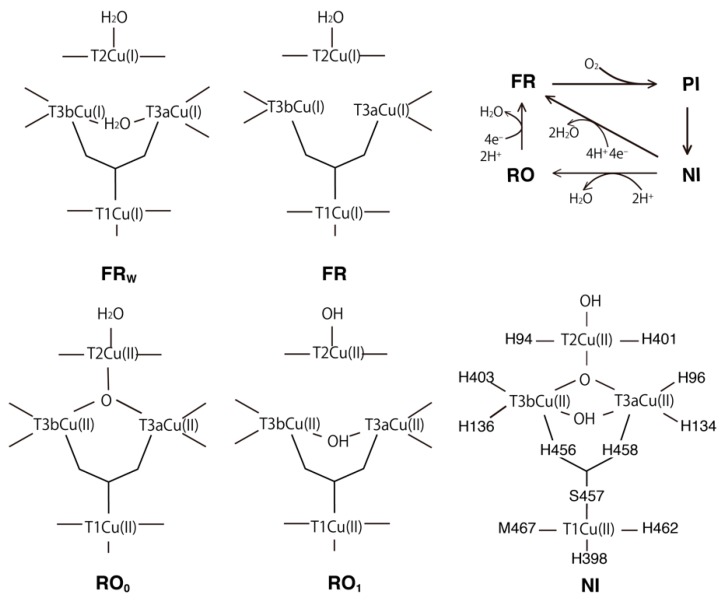
Schematic illustration of the proposed TNC intermediate states of BOD in the entire catalytic cycle. The calculated states were the native intermediate (NI), resting oxidized state in two different protonation states (RO_0_ and RO_1_), and fully reduced states with and without a bridging water molecule (FR_W_ and FR).

**Figure 2 molecules-24-00076-f002:**
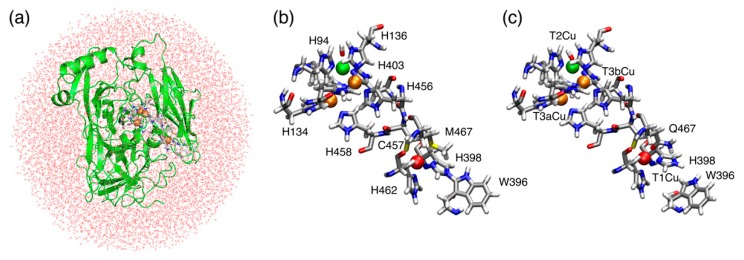
(**a**) Overall view of the QM/MM system used in the present study. (**b**) QM region in the wild-type (WT). (**c**) QM region in the M467Q mutant. The green, orange and red spheres represent the T2Cu, T3Cu and T1Cu atoms, respectively.

**Figure 3 molecules-24-00076-f003:**
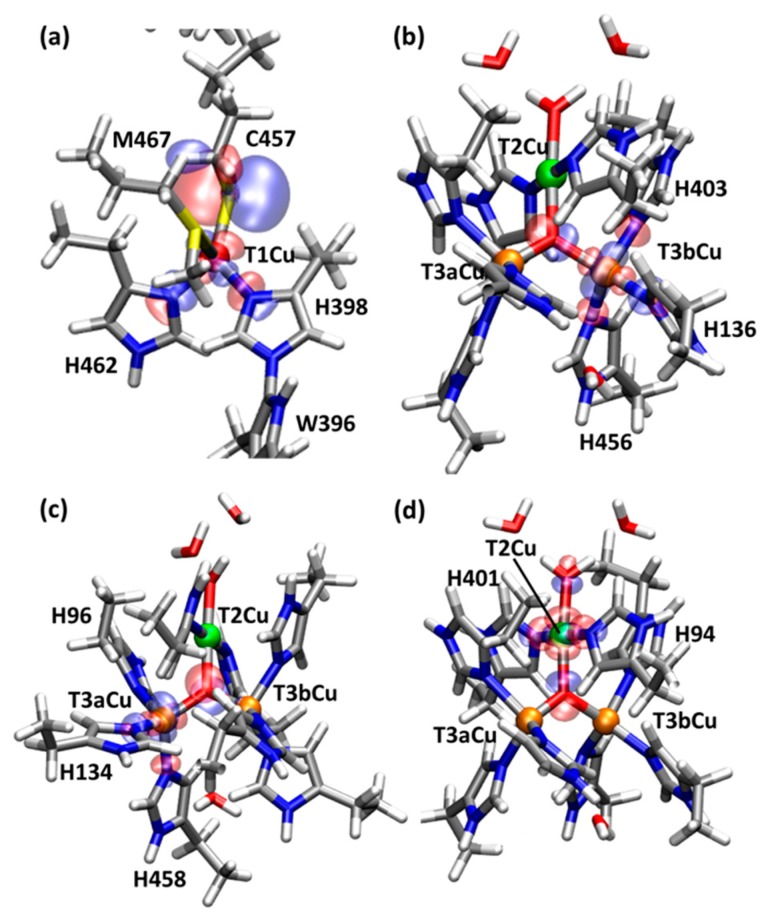
Localized singly occupied natural orbitals (LSONOs) calculated at RO_0_, which contain four d orbitals mainly localizing on each Cu center ((**a**) T1, (**b**) T2, (**c**) T3a and (**d**) T3b). For clarity, the pictures are enlarged only for the part of the orbital distribution.

**Figure 4 molecules-24-00076-f004:**
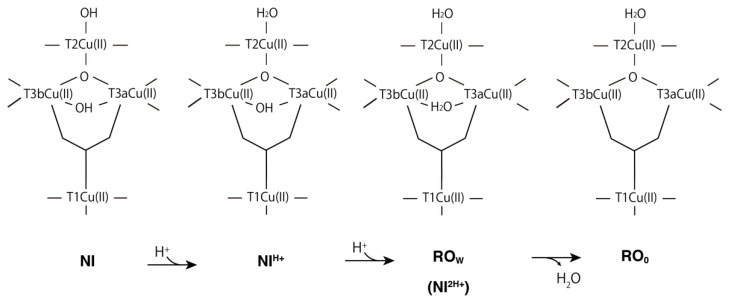
Intermediate states for the NI and RO states. The order of the two-step protonation and the subsequent water elimination is proposed as shown in this scheme. The first and second intermediate states are noted as NI^H+^ and RO_W_, respectively.

**Figure 5 molecules-24-00076-f005:**
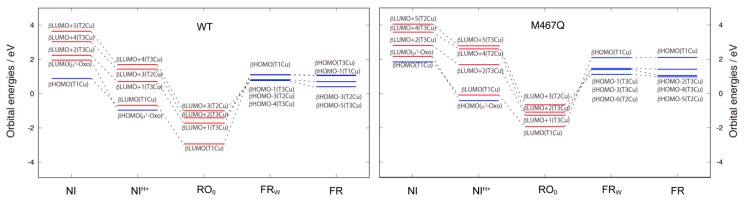
β molecular orbital energies of the WT and M467Q models.

**Figure 6 molecules-24-00076-f006:**
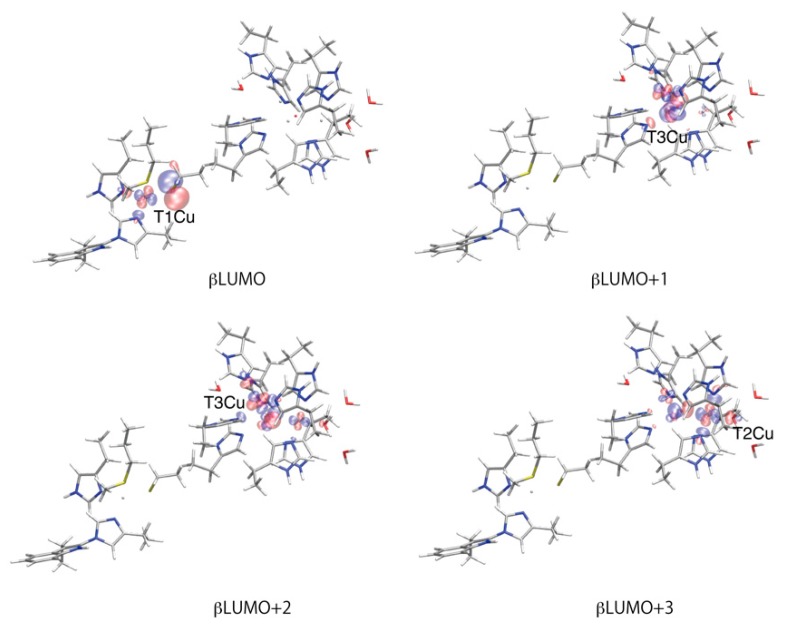
β unoccupied molecular orbitals calculated at the RO_0_ in WT. These MOs are similar to the LSONOs shown in Figure 3, and these MOs are doubly occupied in the FR states.

**Figure 7 molecules-24-00076-f007:**
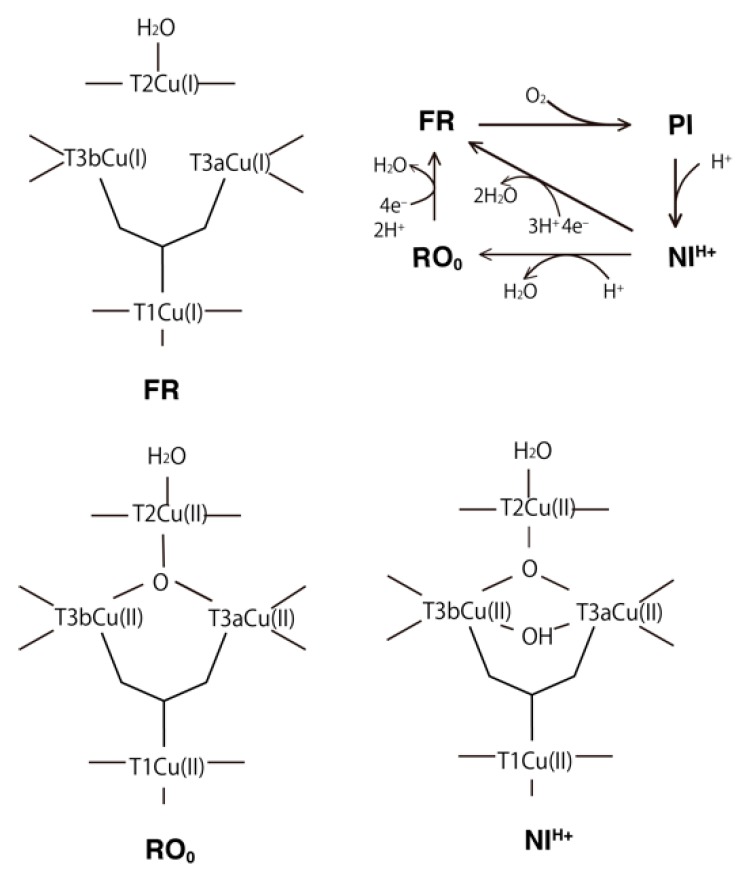
Schematic illustration of the catalytic cycle of BOD based on the present QM/MM calculations.

**Table 1 molecules-24-00076-t001:** Calculated Löwdin atomic spin densities.

	T1Cu	S (C457)	T2Cu	T3aCu	T3bCu	μ^3^-oxo	μ^2^-OH/W
NI	0.00	0.00	0.59	0.64	0.66	1.07	0.28
NI^H+^	0.13	0.30	0.58	0.62	0.65	0.75	0.29
RO_W_	0.30	0.55	0.59	0.46	0.65	0.76	0.08
RO_0_	0.31	0.57	0.62	0.53	0.65	0.63	-
RO_1_	0.32	0.57	0.50	0.56	0.60	-	0.22
FR_W_	0.00	0.00	0.00	0.00	0.00	-	0.00
FR	0.00	0.00	0.00	0.00	0.00	-	-

**Table 2 molecules-24-00076-t002:** Key atomic distances for the TNC (*R*/Å).

	T2Cu-T3aCu	T2Cu-T3bCu	T3aCu-T3bCu	Cu- μ^3^-oxo (T2Cu, T3aCu, T3bCu)	Cu- μ^2^-OH/W (T3aCu, T3bCu)
NI	3.84	3.75	3.14	2.07, 2.18, 2.02	1.94, 1.90 (μ^2^OH)
NI^H+^	3.60	3.56	3.06	1.94, 1.99, 2.07	1.95, 1.90 (μ^2^OH)
RO_W_	3.52	3.59	3.38	2.03, 1.94, 2.11	2.58, 1.99 (μ^2^W)
RO_0_	3.48	3.48	3.27	2.09, 1.94, 1.89	–
RO_1_	4.28	4.21	3.66	–	1.99, 1.93 (μ^2^OH)
FR_W_	4.34	4.52	4.79	–	2.45, 2.46 (μ^2^W)
FR	4.27	4.45	4.60	–	–
2XLL	4.06	4.07	4.87	–	–
6IQZ	4.18	3.98	4.98	–	2.69, 2.29
3UAA	3.65	3.61	3.22	1.83, 2.07, 2.27	2.06, 1.82
3UAE	4.40	3.98	4.96	–	2.77, 2.27

**Table 3 molecules-24-00076-t003:** Similarities (in Å unit) of TNC among all the calculated states and X-ray structures. ^a^

	*l*	RMSD (2XLL)	RMSD (6IQZ)	RMSD (3UAA)	RMSD (3UAE)
NI	3.538	0.716	0.766	0.105	0.749
NI^H+^	3.382	0.720	0.766	0.051	0.765
RO_W_	3.493	0.610	0.654	0.097	0.669
RO_0_	3.405	0.658	0.702	0.088	0.713
RO_1_	4.023	0.566	0.614	0.327	0.579
FR_W_	4.538	0.232	0.245	0.683	0.257
FR	4.433	0.245	0.270	0.606	0.282
2XLL	4.278	0	0.073	0.674	0.147
6IQZ	4.309	0.073	0	0.721	0.202
3UAA	3.477	0.674	0.721	0	0.719
3UAE	4.394	0.147	0.202	0.719	0

**^a^** Underlines for the RMSD emphasize the smallest/closest data sets to the X-ray structures.

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
