# Peer review of "Structural Changes of the Trinuclear Copper Center in Bilirubin Oxidase upon Reduction"

_molecules, 2018, doi:10.3390/molecules24010076_

Round 1
Reviewer 1 Report
This is a fairly straightforward QM/MM study looking at differences in oxidation state of the Cu centers in a metalloenzyme, and trying to draw some conclusions about its catalytic cycle on the basis of this information. The only problem is a lack of detail regarding the methodology. I’m pretty sure that I can guess what was done but I shouldn’t have to guess. Therefore, in the methods section please state clearly (1) how was the initial structure obtained? Presumably via classical MD. Details are needed. (2) Then what? It is never actually stated explicitly *what* QM/MM calculations were performed, although it is vaguely hinted that these are energy minimizations (geometry optimizations) rather than MD. Details are needed once again, along with some justification as for why this is adequate. If it really is just energy minimization rather than MD, why is it reasonable to look simply at minimum-energy structures? These issues need to be fully addressed, and to do so will require more than a one-paragraph “Methods” section. I think this material deserves to be moved forward in the text, between the Intro and the Discussion. I don’t like the fact that details such as functional, basis set, etc. are left to the end. Basically, it connotes a point of view where the authors expect the reader to assume that the methodology is correct and just read the discussion, which is the wrong way to read a computational paper in my opinion. There are enough ways to go wrong that the details should be presented *first*, because that may very well color how I interpret the results.
Since insufficient details were provided to be certain about the computational procedure, this needs to be reviewed again once those details have been clarified.
Author Response
Thank you very much for your suggestion. According to your suggestion, the computational details section moved in section 2. And details for the computational procedures for the initial structure setting are added in the computational details section.
Our reply for the question, why MD calculations were not performed, is that BOD contains Cu atoms and the protonation states were undetermined. Therefore, proper force field, especially for the TNC cannot be constructed. Another reason is that we can use the series of high resolution X-ray structures of BOD with less than 1.6 Angström resolution, we can trust the atomic coordinates for heavy atoms. For the solvent water molecules and hydrogen atoms, which are added in our calculations, they were all annealed in the initial setting procedure. As discussed in the main text, structural changes upon the reduction were very small, ~0.5 Angström mainly for the Cu-Cu distances in TNC, and these structural changes can be easily buried if the positions of the amino acid residues coordinating to the TNC are largely perturbed after long thermal MD. Therefore, in our present QM/MM approach, even though the very local structural changes can be detected by the energy minimization calculations, we constructed the most adequate and realistic theoretical model as much as possible. We want to emphasize that the large QM region was adapted, this means that the electronic structure is accurate by minimizing the artificial effect from the terminal H-atom link.
Reviewer 2 Report
Report on the paper entitled "Structural Changes of the Trinuclear Copper Center in Bilirubin Oxidase upon the Reduction", by Takaki Tokiwa, Mitsuo Shoji, Vladimir Sladek, Naoki Shibata, Yoshiki Higuchi, Kunishige Kataoka, Takeshi Sakurai, Yasuteru Shigeta, and Fuminori Misaizu, submitted to Molecules.
This is a reasonable study of the changes in the structure of the trinuclear copper center in bilirubin oxidase through reduction. A QM/MM approach is used for the purpose, with standard parameters (B3LYP-D3 functional and LANL2dz basis sets for the QM part, which contains, besides the 4 Cu atoms, 13 residues, and Amber force field for the MM environment. The size of the model is definitely large enough to let trust in the calculation reliability. The paper is clearly written, and I think it may be published in the present form.
Author Response
Thank you very much for your review comments.
Round 2
Reviewer 1 Report
The computational methods are now adequately described, which was my only criticism of the original version of the work.